# Optoelectronic Effects of Copper–Indium–Gallium–Sulfur (CIGS_2_)-Solar Cells Prepared by Three-Stage Co-Evaporation Process Technology

**DOI:** 10.3390/mi14091709

**Published:** 2023-08-31

**Authors:** Tzu-Chien Li, Chia-Wen Chang, Chia-Chun Tai, Jyh-Jier Ho, Tung-Po Hsieh, Yung-Tsung Liu, Tsung-Lin Lu

**Affiliations:** 1Department of Electrical Engineering, National Taiwan Ocean University, No. 2, Peining Rd., Keelung 20224, Taiwan; 2Photovoltaic Technology Division, Green Energy and Environment Research Laboratories, Industrial Tech Research Institute, No. 195, Sec. 4, Chung-Hsing Rd., Chutung, Hsinchu 310401, Taiwan

**Keywords:** Cu-(In,Ga)-S_2_ (CIGS_2_) solar cells, three-stage co-evaporation technique, [Cu]/([Ga] + [In]) (CGI)-ratio stoichiometry, opto-electric sensing performance, eco-friendly community

## Abstract

In this paper, the performance of Cu-(In,Ga)-S_2_ (CIGS_2_) solar cells with adjusting composite [Cu]/([Ga] + [In]) (CGI)-ratio absorber was explored and compared through an improved three-stage co-evaporation technique. For co-evaporating CIGS_2_ absorber as a less toxic alternative to Cd-containing film, we analyzed the effect of the CGI-ratio stoichiometry and crystallinity, and explored its opto-electric sensing characteristic of individual solar cell. The results of this research signified the potential of high-performance CIGS_2_-absorption solar cells for photovoltaic (PV)-module industrial applications. For the optimal CIGS_2_-absorption film (CGI = 0.95), the Raman main-phase signal (A1) falls at 291 cm^−1^, which was excited by the 532 nm line of Ar^+^-laser. Using photo-luminescence (PL) spectroscopy, the corresponding main-peak bandgaps measured was 1.59 eV at the same CGI-ratio film. Meanwhile, the best conversion efficiency (*η* = 3.212%) and the average external quantum efficiency (EQE = 51.1% in the visible-wavelength region) of photo-electric properties were achieved for the developed CIGS_2_-solar cells (CGI = 0.95). The discoveries of this CIGS_2_-absorption PV research provided a new scientific understanding of solar cells. Moreover, this research undeniably contributes to a major advancement towards practical PV-module applications and can help more to build an eco-friendly community.

## 1. Introduction

In recent years, photovoltaic (PV) technologies have been the most abundant form of renewable energy of a fast-growing industry [1]. For many PV technologies, solar cells with thin-film light trapping structures have attracted extensive attention due to their advantages such as nanowires for light weight, and manufacturing flexibility [2,3,4]. Amongst thin-film PV cells with a copper–indium–gallium–selenide (CIGSe) absorber [5,6,7], a low-temperature fabrication process can be applied to a glass or flexible polyimide (PI) substrate for building-integrated industry PV applications [8]. However, when the substrate temperature is below 500 °C, alkali elemental (Na) diffusion from the substrate is limited for the high-efficiency CIGSe solar cells [9].

In order to solve this limitation, a three-stage process with Na postdeposition treatment (PDT) was reported [10], thus improving the crystallinity and the grain size of the CIGSe absorption layer deposited for the low-temperature process. For the design of CIGS_2_-absorber thin-film solar cells on flexible substrates, we focused on developing the novelty of the modified three-stage co-evaporation process, in the meantime maintaining good interface characteristics between the buffer layer and the CIGS_2_ absorber on the tunable bandgap effect [4,5,6,7,8]. ZnO thin film was deposited on the CdS surface through the chemical bath deposition (CBD) process to form a stack of buffer layers, denoted as i-ZnO/CdS.

However, the impact on environmental issues and human health has recently been regarded as one of the top priorities. Therefore, developing alternatives to highly toxic cadmium (Cd)-containing materials has become a primary issue in the eco-friendly research. Thus, a copper–indium–gallium–sulfur (CIGS_2_) can be used as a less toxic alternative to Cd-containing semiconductors. Moreover, CIGS_2_ is a promising absorber material for the fabrication of high-efficiency thin-film solar cells thanks to its well-adapted bandgap (approximately 1.5 eV) [11,12]. A gradient in the Ga distribution has also been found, and CIGS_2_ was also investigated to be an absorber for the top cell of the tandem configuration [13].

For the CIGS_2_-solar cell fabrication in this study, the absorber layer was prepared by a modified three-stage sequential co-evaporation method [9], which can effectively adjust the ratio of the [Cu]/([Ga] + [In]) (CGI) process. In this approach, combined with an additional Cu-rich for composite-ratio (CGI ratio for Cu-contents of Cu/(In + Ga)) deposition phase, an annealing process was utilized by different process duration after stage 2. Such a method has shown tremendous potential to improve the crystallinity of the CIGS_2_ films deposited at low temperatures. 

At the same time, X-ray diffraction (XRD) analysis, Raman-shift spectroscopy, and photo-luminescence (PL) spectroscopy were used to study the crystallization characteristics of the CIGS_2_ absorber layer with different CGI ratios. The two main objectives of this experiment were to increase conversion efficiency (*η*) and to improve the external quantum efficiency (EQE) in the visible-wavelength region, all for CGI ratio (composite-ratio polymer) approaches 1 of the developed CIGS_2_-solar cells. Under the premise of environmental protection for promoting the commercialization of these technologies, we expected to diminish the manufacturing temperature, lessen the toxic material, and reduce the production cost. These eco-friendly cells could be effectively applied to mass production for commercial PV-module applications.

## 2. Experiments and Measurements

The experimental process, as described in our previous work [6], mainly used the co-evaporation method to make the absorption layer, the CBD process, and sputtering method to plate the buffer layer and the barrier layer, and finally used the sputtering method to make the Ni-Al surface electrode. For the modified three-stage co-evaporation method in this study, four evaporation sources including Cu, In, Ga, and S were deposited on the substrate as the absorption layer. Table 1 illustrates the deposition parameters of the CIGS_2_ thin-film absorption layer for preparing sample S1, S2, and S3 with different CGI ratios [14], changing from 0.67 to 1.96 for better crystal thickness and less secondary phase signal. All the parameters were varied by the substrate temperature (°C) and the evaporation time (minutes) of adding the evaporation source (✓ check marked) during each stage co-evaporation process.

To investigate the characteristics for the three samples with different absorption layers, Table 2 summarizes the surface composition and the corresponding images of energy-dispersive X-ray spectroscopy (EDS) spectrum of their respective absorption layers. The CGI ratios of sample S1, S2, and S3 were separated for 0.78, 0.95, and 1.25. Meanwhile, the sample S2 ([Ga]/([Ga] + [In]), GGI = 0.10) of this table confirmed that CIGS_2_ is a high-efficiency device, and the content of Ga atoms (between 0.1 and 0.3) to the GGI in Ga [14]. The crystal structure of films was evaluated by a Rigaku X-ray diffraction (XRD) with Ni-filtered Cu Ka radiation. An atomic force microscope (AFM) instrument (Bruker INNOVA SPM) was used to scan the surface terrain in a typical area of 2.5 × 2.5 mm^2^ (512 × 512 pixels) of the CGIS_2_ film on a vibrating-free platform. The root mean square (RMS) surface-roughness values were obtained using the software that came with the instrument. Furthermore, the optical and electrical properties of our cells were also measured using commercially available systems: scanning electron microscopy (SEM) and EDS (Model: JSM-6500F, JEOL Ltd.), micro-Raman scattering (UniDRON, CL Technology Co.) incorporated with microwave-induced photo-conductance decay (μ-PCD) system (Model: U-2001), PV conversion efficiency measuring system (model: Oriel-91192/AM 1.5 GMM), and the EQE measurements (model: QE-R, Enlitech).

## 3. Results and Discussion

Figure 1 demonstrates top-view and cross-section SEM images of CIGS_2_ absorption layers for different CGI ratios. In all cases, the surface morphology was dominated by the underlying granular structure of the CIGS_2_, which remained visible beneath the over-layers. It was seen that the grain size of the Cu-poor (sample S1) absorber (0.9 μm of left of Figure 1a) was slightly smaller compared with Cu-rich (sample S3) absorber (1.3 μm of left of Figure 1c). Meanwhile, the thickness of Cu-poor absorber (1.875 μm of right of Figure 1a) was thicker than that of Cu-rich absorber (1.547 μm of right of Figure 1c). Moreover, quantization from the EDS spectra of CIGS_2_ films illustrated the CGI ratios of 0.78, 0.95, and 1.25 for samples S1, S2, and S3, respectively (shown as Figure 1). This led to a slower reaction as well as growth rate leading to the formation of uniform nanoparticles [15]. Furthermore, the incorporation of the Cu-rich (samples S2 and S3) of CIGS_2_-absorption layer can distinctly boost grain growth and reduce the fine-grain layer, thus greatly improving the absorber crystallinity and reducing the trap state density [16]. Table 3 plots for 2-D and 3-D AFM images of as deposited different CGI-ratio samples, in which *R*_a_ and *R*_max_ stand for the average centre-line and the maximum surface-height roughness, respectively. As is evident from the S2 image’s relatively compact and flat surface structure, the film was composed of cone-shaped columns randomly distributed over the surface of the film. Thus, the films’ surface was rough and porous with RMS surface roughness (*R*_q_) of 97.3 nm. These results were consistent with the results in the literature [16,17], indicating that a more porous CGI-ratio film structure led to a greater grain size, as seen in Figure 1.

Figure 2 shows XRD patterns (under 5° incident angle) of CIGS_2_ layers with different CGI ratios, in which the chalcopyrite structure of these films was determined by XRD using Cu-K radiation of 1.5418 Å. The peak pattern was in good agreement with the Joint Committee on Powder Diffraction Standards (JCPDS) reference diagrams for the corresponding bulk phases. The (112), (204/220), and (116/312) lattice plane, respectively, located at 27.9°, 46.5°, and 55.0° (2-theta values) from CuInS_2_ (JCPDS # 00-042-1475). The secondary-phase signal was located at CuS (103) at 32.38°, Mo (110) at 40.5°, MoS_2_ (104) at 44°, and CuInS_2_ (440) at 48.06°.

Figure 3 illustrates the Raman spectra of CIGS_2_ thin-film absorption layers, which exhibited peaks that can be attributed to the phonon modes of the ternary CuInS_2_ chalcopyrite. The absorption layers were excited by the 532 nm line of an argon (Ar^+^) laser at room temperature. Meanwhile, the modes of secondary phase were formed during the deposition process. The most peak at 291 cm^−1^ was assigned to the A1-mode of CuInS_2_, thus coinciding a noticeable contribution from Cu–Au mode (310 cm^−1^) in CIGS_2_ films [18].

At same modulation source as Raman spectra on CIGS_2_ thin films, Figure 4 plots the energy bandgap (eV)—the photoluminescence (PL) spectra measured at the 600–980 nm wavelength transmission. Therein, the Cu-rich with samples S2 and S3, the main peak displaced to about 1.59 and 1.61 eV, which was possibly induced by the secondary phase of CuS (103) at 32.38° (shown in XRD pattern of Figure 2) [16]. In the lowest CGI-ratio film (sample S1), it did not have an obvious peak. The samples S2 and S3 showed the obvious and wanted peak at 1.59 and 1.61 eV, respectively. However, sample S1 had the unwanted peak at 1.41 eV of CuIn_5_S_8_, thus increasing defect density of the CIGS_2_ film in concurrence with the literature reports [19].

The current density-voltage (*J*–*V*) characteristics of solar cells were measured under illumination and under the conditions of AM1.5G and 1000 W/cm^2^. Figure 5 shows the *J*–*V* curve under illumination and the electrical properties of the CGIS_2_-layer solar cells are summarized on the top. The best performance of sample S2 (CGI = 0.95) yielded the highest efficiency of the CGIS_2_-layer solar cell, with short-circuit current (*J*_SC_) = 15.53 mA/cm^2^, open-circuit voltage (*V*_OC_) =590 mV, series resistance (*R*_S_) = 189 Ω, and conversion efficiency (*η*) = 3.212%, yielding a gain of 2.304%absolute compared to the S1 cell. Although there was improvement in fill factor (*F*.*F*.) for the higher CGI-ratio solar cell, it resulted in larger-*J*_SC_ value. Upon further investigation, it appeared that the cause for the lower CGI-ratio device required with this synthesis technique (CGI ratio ~1.0), which showed an approximate 1.6-eV-bandgap phenomena that was also observed with the CuGaS_2_ absorption layer at the PL-spectra of Figure 4, as described previously, thus improving optoelectronic quality and increasing the charge-carrier lifetime of the absorption layer [20]. 

Figure 6 shows the EQE effect for our CIGS_2_-absorption solar cells with different CGI ratios. In this study, for the visible-wavelength range (400–700 nm), the average EQE value of our CIGS_2_-solar cell was increased as the CGI-ratio was increased, due to the low transmittance of the absorption layer [21]. This was attributed to the light absorption effect [6]—the absorber structure (CIGS_2_ layer) adopted narrower bandgap of CIGS_2_ (~1.61 eV), thus avoiding more incident light into the main absorption layer. This phenomenon also indicates that, as shown in Figure 5, only a very small number of carriers were generated in the lower CGI-ratio absorption layer after illumination, resulting in a decrease in *J*_SC_ and *η* values. Additionally, it can be noticed that in the visible-wavelength range of Figure 6, sample S3 exhibited a great improvement in EQE as compared with sample S1 by approximately 29.8%absolute. This was presumed to be due to the inability of the absorber layer to efficiently extract current in the visible-wavelength region.

The solar-panel manufacturing process brought together six different CIGS_2_-absorber cells to create a functioning PV-array module, the size of which was the dimension of 30 × 30-cm^2^ area. For commercial applications of mass production of PV modules, the overall performance will vary significantly under different photo-intensity conditions, which will have a serious impact on the yield of PV systems. Variations in the intensity of solar radiation falling on a PV module affect many of its parameters, including *V*_OC_, *F*.*F*., conversion efficiency, and output power. For a PV module (CGI = 0.95) prepared with the optimal CIGS_2_-solar cells under photo intensity set at 1 KW/m^2^, Figure 7 plots the current (A)-voltage (mV) cures of a module with varying ambient temperatures. Among them, the *V*_OC_ value decreased (from 570 to 410 mV) with the increase in the ambient temperature (from 30 to 75 °C). At these ambient temperatures, the corresponding optoelectrical performance (*V*_OC_, *F*.*F*., and *η* values) of the developed PV module was attached on the top of Figure 7. This was the negative-temperature effect of *V*_OC_ value, which, in turn, led to a drop in its *η* value (from 5.47% to 3.21%) with the same point of view found in [22].

As photo intensity increased, again, the incident photo energy was absorbed more because a greater percentage of the incident light had enough energy to raise charge carriers from the valence band to the conduction band [23]. For a PV module (CGI = 0.95), Figure 8 illustrates the output power (W) relation with voltage (V), and it showed a linear upward trend with the gradual increase in photo intensity (from 250 to 1000 W/m^2^). On the contrary, this was the positive-photo-intensity effect of *V*_OC_ value, which, in turn, led to a raise in its *η* value (from 1.58% to 5.73%), which was consistent with the findings in [23].

## 4. Conclusions

In summary, the modified three-stage co-evaporation method was used to prepare the CIGS_2_-absorber solar cell by adjusting the CGI-ratio to obtain the optimal PV characteristics. All the CIGS_2_-absorption layers were characterized by the EDS, XRD, Raman, and PL spectra, which were persuasive and distinguished the effect of CGI ratios on CIGS_2_ thin-film layers. The SEM and AFM images (Figure 1 and Table 3) clearly showed nanocrystalline CIGS_2_-absorber solar-cell structures without interfacial reactions, and these thick interfaces between the layers indicated that all layers are physically stable. For the developed CIGS_2_-absorption film (CGI = 0.95), the Raman main-phase signal (A1) fell at 291 cm^−1^, which was excited by the 532 nm line of Ar^+^-laser. Using PL spectroscopy, the corresponding main-peak bandgaps measured was 1.59 eV at the same CGI-ratio film. Meanwhile, the best conversion efficiency (*η* = 3.212%) and the average EQE (=51.1% in the visible-wavelength region) of photo-electric properties were achieved for the developed CIGS_2_-solar cells (CGI = 0.95).

For PV modules used in commercial mass production, the overall performance will vary greatly under different photo intensities. Variations in the intensity of solar radiation falling on a PV module affect many of its parameters, including *V*_OC_, *F*.*F*., *η*, and output power. For a PV module (CGI = 0.95) prepared with the optimal CIGS_2_-solar cell, the *η* values decreased with rising ambient temperature (30~75 °C). At the same time, its output powers also increased with increasing photo intensity. In addition, these optimal CIGS_2_-absorption layers showed excellent thermal stability at all ambient temperatures under photo intensity set at 1 KW/m^2^. On the premise of improving the performance of the developed devices, we hope that the proposed technology can not only reduce the process temperature and production cost but also lessen the toxic Cd-containing materials. Except the modified three-stage co-evaporation method in this study, the novel discoveries of this CIGS_2_-absorber research provide a new scientific understanding of PV applications. From a perspective of sustainability environmental consciousness, thus, the eco-friendly PV modules can be effectively applied to commercial mass production.

## Figures and Tables

**Figure 1 micromachines-14-01709-f001:**
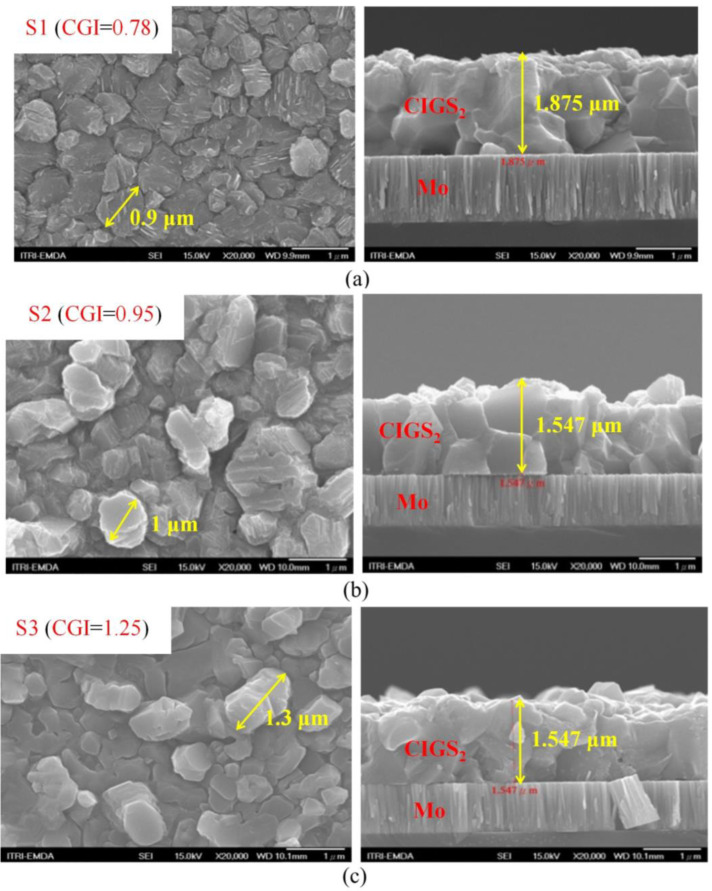
The top view (**left**) and cross-sectional view (**right**) of SEM images with the CIGS_2_ absorber layer for the CGI ratios are: (**a**) CGI = 0.78 (Cu-poor), (**b**) CGI = 0.95, and (**c**) CGI = 1.25 (Cu-rich), respectively. The grain size (from 0.9 to 1.3 μm) increases with copper content for crystal thickness in the range of 1.547~1.875-μm.

**Figure 2 micromachines-14-01709-f002:**
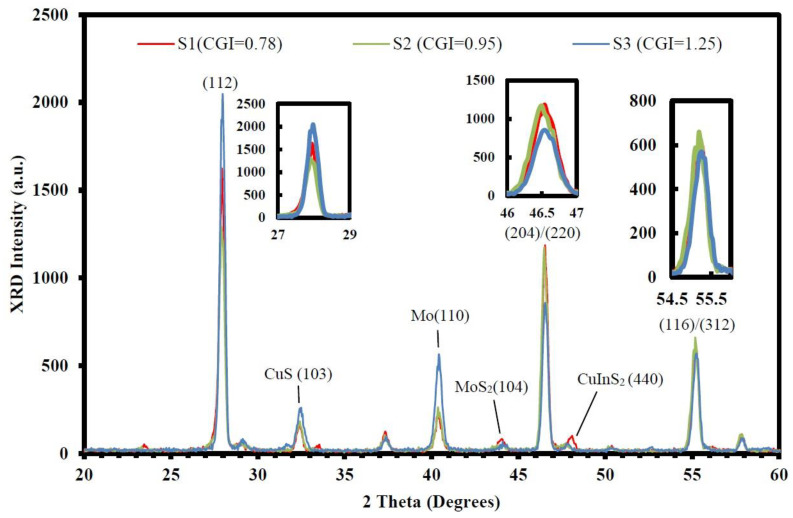
The X-ray diffractograms (under the 5° incident angle) are mainly in the deeper part of the CIGS_2_ absorption layer with different CGI ratios. The 2θ values of the main diffraction peaks appear at 27.9°, 46.5°, and 55°from zoom-in patterns (inserted images), respectively, on (112), (204/220), and (116/312) facets. The secondary-phase signal is located at CuS (103) at 32.38°, Mo (110) at 40.5°, MoS_2_ (104) at 44°, and CuInS_2_ (440) at 48.06°.

**Figure 3 micromachines-14-01709-f003:**
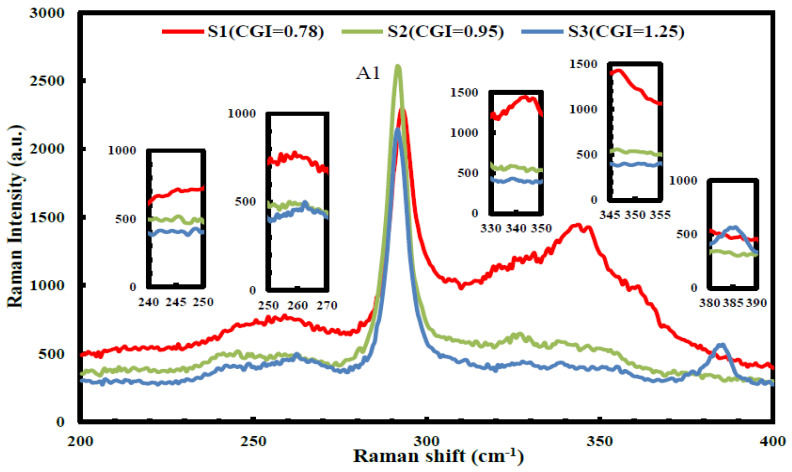
The Raman spectra of CIGS thin film absorption layer prepared with different sample of S1, S2, and S3, which contain various concentrations of Cu-Au ordering, in which the Raman main-phase signal (A1) falls at 291 cm^−1^, which is excited by the 532 nm line of Ar^+^-laser.

**Figure 4 micromachines-14-01709-f004:**
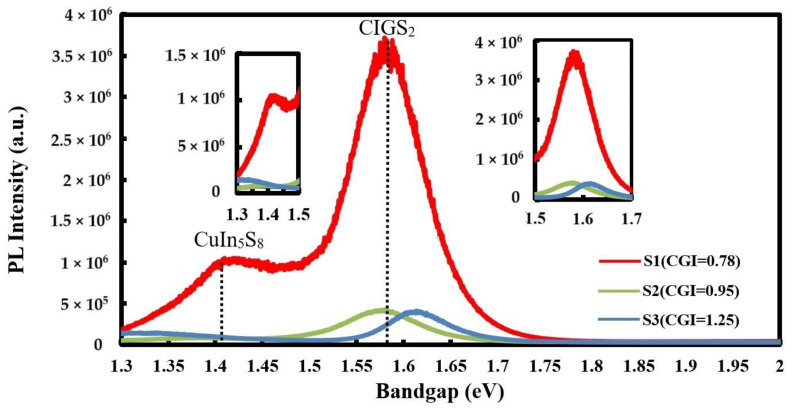
The photoluminescence (PL) spectra of CIGS_2_ thin film absorption layer, which the main-peak bandgaps measured are 1.58, 1.59, and 1.61 eV, respectively, for S1, S2, and S3 samples.

**Figure 5 micromachines-14-01709-f005:**
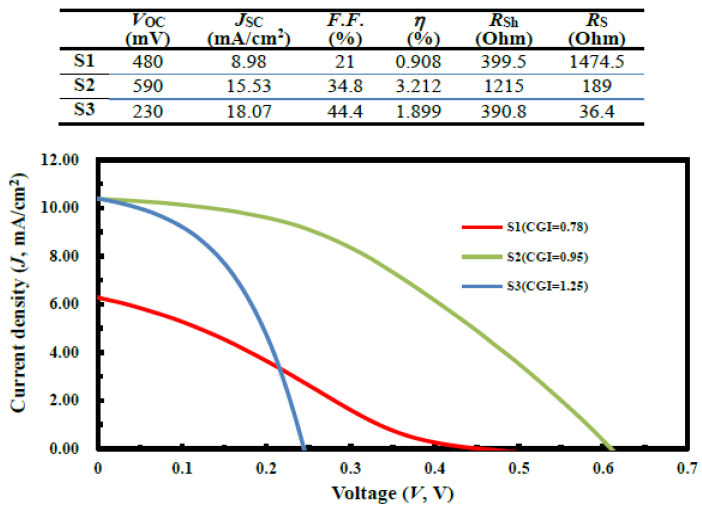
The current density-output voltage (*J*–*V*) curves prepared from different samples of S1, S2, and S3, where the optoelectrical performance of CIGS_2_-solar cells has been attached on the top.

**Figure 6 micromachines-14-01709-f006:**
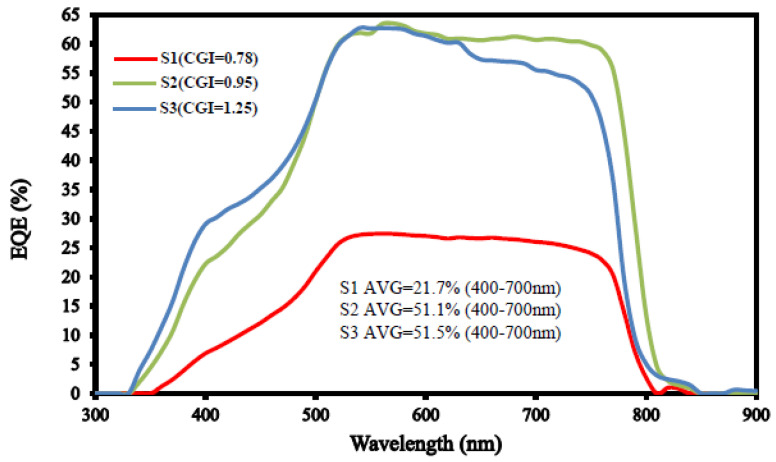
The EQE of CIGS_2_-solar cells prepared with different samples of S1, S2, and S3. The S3 sample EQE value (average for the wavelength range of 400–700 nm) is about 51.5% higher than that of the other samples.

**Figure 7 micromachines-14-01709-f007:**
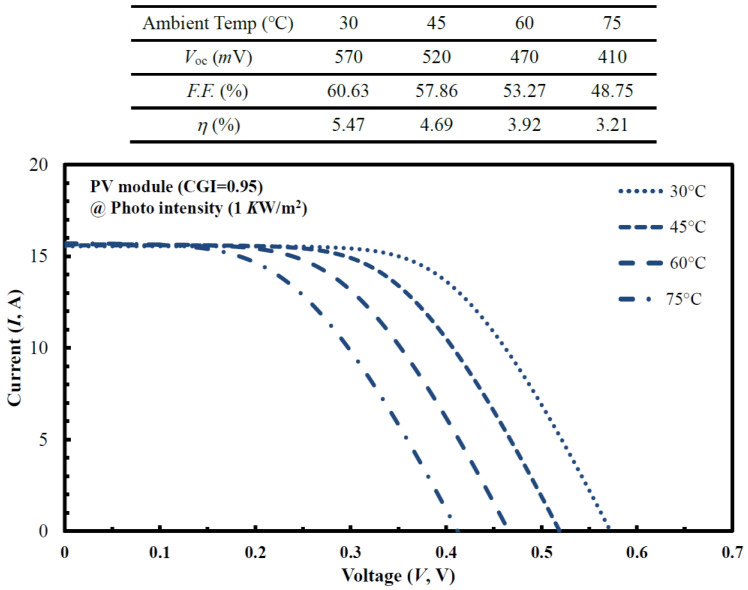
Under the fixed photo intensity (KW/m^2^) for the optimal CIGS_2_-solar cells prepared with a PV module (CGI = 0.95), the current-voltage (I–V) curves observed from different ambient temperatures (30~75 °C).

**Figure 8 micromachines-14-01709-f008:**
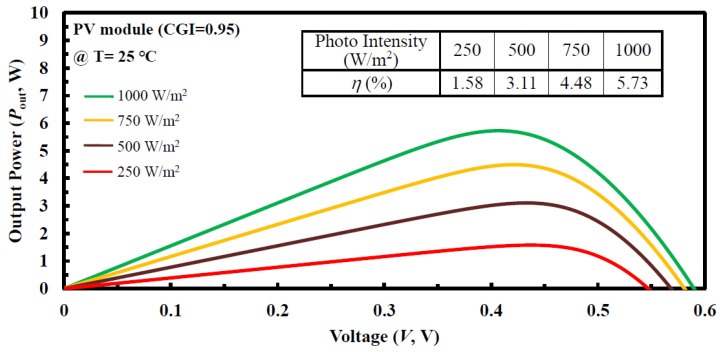
Under the fixed ambient temperature (25 °C) for the optimal CIGS_2_-solar cells prepared with a PV module (CGI = 0.95), the output power-voltage (*P*_out_-V) curves observed from variable photo intensity (250~1000 W/m^2^).

**Table 1 micromachines-14-01709-t001:** Experimental parameters of three-stage co-evaporation deposition for CIGS_2_ thin-film absorption layer prepared with different CGI ratios, (***a***) S1 (CGI = 0.78), (***b***) S2 (0.95), and (***c***) S3 (CGI = 1.25).

**(*a*) S1 (CGI = 0.78)**	**Cu**	**In**	**Ga**	**S**	**Temp (°C)**	**Time (min)**
Stage 1	✓	✓	✓	✓	380	48.5
Stage 2				✓	380–640	40
Stage 3		✓	✓	✓	640	20
**(*b*) S2 (CGI = 0.95)**	**Cu**	**In**	**Ga**	**S**	**Temp (°C)**	**Time (min)**
Stage 1	✓	✓	✓	✓	380	41
Stage 2	✓			✓	380–640	31
Stage 3		✓	✓	✓	640	16
**(*c*) S3 (CGI = 1.25)**	**Cu**	**In**	**Ga**	**S**	**Temp (°C)**	**Time (min)**
Stage 1	✓	✓	✓	✓	380	37
Stage 2	✓			✓	380–640	33
Stage 3		✓	✓	✓	640	9

**Table 2 micromachines-14-01709-t002:** The surface composition and the corresponding images (below) of EDS spectrum of CIGS_2_ thin-film absorption layer prepared with different CGI ratios, S1, S2, and S3 for CGI = 0.78, CGI = 0.95 and CGI = 1.25, respectively.

Atomic Ratio (%)	Cu K	In L	Ga K	S K	Mo L	CGI	GGI
S1	21.12	24.63	2.43	49.03	2.79	0.78	0.08
S2	24.33	23.05	2.43	46.6	3.59	0.95	0.10
S3	27.21	20.68	0.98	48.75	2.38	1.25	0.04
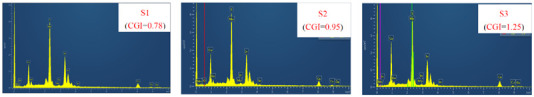

**Table 3 micromachines-14-01709-t003:** The AFM roughness and the corresponding images (below) of CIGS_2_ thin-film absorption layer prepared with RMS (Rq) values, S1, S2, and S3 for 65.2, 97.3, and 130 nm, respectively.

	RMS (Rq)	Ra	Rmax
S1 (CGI = 0.78)	65.2 nm	50.4 nm	407 nm
S2 (CGI = 0.95)	97.3 nm	73.4 nm	483 nm
S3 (CGI = 1.25)	130 nm	104 nm	785 nm
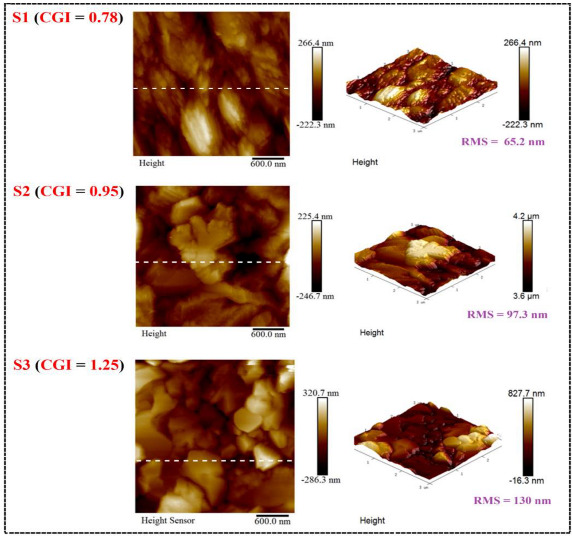			

## Data Availability

Not applicable.

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
