# Peer review of "Optoelectronic Effects of Copper–Indium–Gallium–Sulfur (CIGS2)-Solar Cells Prepared by Three-Stage Co-Evaporation Process Technology"

_micromachines, 2023, doi:10.3390/mi14091709_

Round 1
Reviewer 1 Report
In this paper, the performance of Cu-(In,Ga)-S2 (CIGS2) solar cells with adjusting composite CGI-ratio absorber is explored and compared.
The paper is of very low quality and should be rejected:
1. (Lines 18, 164): It must be cm-1, but no cm-1. In all paper, the degrees, indices and grades must be carefully checked and revised.
2. (Line 21): What is the parameter in the first bracket?
3. (Line 61): There are empty brackets ( ).
4. Data for S1, S2, S3 in Lines 72-73 are repeated in Line 79.
5. (Line 101): “…the EDS spectra of CIGS2 films illustrates the CGI ratios…” Where are shown these spectra?
6. Location of Tables and Figures after main text is not applicable.
7. Table III: What do the parameters Ra and Rmax mean?
8. (Line 106, 107): “Table III plots for 2-D and 3-D AFM images…” Where are these images?
9. (Lines 115, 116): m – What is the physical unit?
10. Fig. 2. There are extraneous icons.
11. What is the Table between Lines 211, 212?
12. (Lines 223, 224): “…as shown in Fig. 5, only a very small number of carriers are generated in the lower CGI-ratio absorption layer after illumination…” Where is it shown in Fig. 5?
13. What is the Table between Lines 261, 262?
14. List of References break the MDPI standards for references.
15. Figures in pp. 12, 14. What is it?
Author Response
Dear All reviewers/Editor & Ms. Catherine Yuan,
Thanks for your e-mail dated at Aug. 11, 2023.
Enclosed please find manuscript files of the paper entitled above, by T.-Z. Li et.al. In the revised one (R1micromachines-2535924Revised.doc), all the comments of the reviewers have been overcome and marked with bold-highlight words. Hope this revised one can be accepted to publish on the micromachines journal (MDPI).
In addition, the special revisions per reviewers’ comments (AnswerQueries(micromachines 2535924-R1).pdf, Please write down "Please see the attachment." in the box) has been attached for editors’ and reviewers’ convenience to check Your kind assistance in dealing with this matter is my most appreciated.
Best regards from Sincerely Yours,
Jyh-Jier HO, Ph.D.

Reviewer 2 Report
The authors employ the experiment method with apparent authority and the results seem valid. manuscript requires improvement
1-The reference cited in introduction section should be improved (1) A few relevant references need to be cited in this article to enrich the background about light trapping structures like, silicon nanowires. e.g., "Effect of the doping concentration on the properties of silicon nanowires." Physica E: Low-dimensional Systems and Nanostructures 56 (2014): 427-430 and "Effect of rapid oxidation on optical and electrical properties of silicon nanowires obtained by chemical etching." The European Physical Journal-Applied Physics 58.2 (2012): 20103
2-The authors should compare the results with other systems the authors can refer to (or cite) this article for reference: "Reduction of absorption loss in multicrystalline silicon via combination of mechanical grooving and porous silicon." physica status solidi c 8.3 (2011): 883-886
3 -Device fabrication section what is the most dangerous chemical element
4- Internal quantum efficiency measurements of the solar cells should be presented to assure that anti-reflection coating variations are meaningful
5- What about the reflectivity, Please add it in the revised paper
6-In the conclusion, almost no perspectives are given for this work. Please develop.
7-What about the stability of these devices in such configuration. This point should be discussed.
8-What about the passivation effect of these devices
9- No information about the cost of this procces
10-The authors should give much more information about the novelty of this paper.
Author Response
Dear All reviewers/Editor & Ms. Catherine Yuan,
Thanks for your e-mail dated at Aug. 11, 2023.
Enclosed please find manuscript files of the paper entitled above, by T.-Z. Li et.al. In the revised one (R1micromachines-2535924Revised.doc), all the comments of the reviewers have been overcome and marked with bold-highlight words. Hope this revised one can be accepted to publish on the micromachines journal (MDPI).
In addition, the special revisions per reviewers’ comments (AnswerQueries(micromachines 2535924-R1).pdf) has been attached for editors’ and reviewers’ convenience to check Your kind assistance in dealing with this matter is my most appreciated.
Best regards from Sincerely Yours,
Jyh-Jier HO, Ph.D.

Reviewer 3 Report
General comment:
In this manuscript, the authors presented a study of optoelectronic effects of CIGS solar cells prepared by three-stage co-evaporation process technology. This work is interesting. Accordingly, I would like to recommend this article. Selected comments go as follows.
Comment 1:
The authors should compare the current study with the others to show the novelty in the introduction.
Comment 2:
The authors should provide the first use of an abbreviation immediately before or after the expanded form. For instance, CGI, GGI…
Comment 3:
CIGS2 or CIGS2? The authors should make it consistent in the manuscript.
Comment 4:
In Fig. 3, the peak related to A1-mode of CuInS2 in S1 or S3 is shifted as compared to that in S2. The authors should explain it.
Comment 5:
The authors mentioned the secondary phase of CuS in Fig. 4. Can it be observed in Raman (Fig. 3)?
Comment 6:
The author should mention the process of solar module in the experimental part. What is the size of solar module?
Comment 7:
In Fig. 7, the Voc is decreased with the increased temperature. The author should explain it.
Author Response

(The authors gave the same response as above.)

Reviewer 4 Report
This paper explores and compares the performance of Cu-(In,Ga)-S2 (CIGS2) solar cells with adjusting composite CGI-ratio absorbers through an improved three-stage co-evaporation technique, and obtaining some valuable results. However, there are many serious issues needed to be improved.
1. The ratios of different CGI are irregular (0.78, 0.95, and 1.25), the author should explain the reasons.
2. The captions of Figure 3 and Figure 4 are missing, which is a serious attitude issue. Which color is S1, S2 S3?
3. The captions of Figure 5 and Figure 5 are incorrect ( the figure is no grey color for S2).
4. For PV modules, stability is crucial. Thus, the change in PCE of devices over time is indispensable, rather than ambient temperatures and light intensity. The testing of device stability in this article is meaningless and unrepresentative.
Moderate editing of English language required
Author Response

(The authors gave the same response as above.)

Round 2
Reviewer 1 Report
The authors have performed necessary revision of the paper. It can be accepted in the present form.
Author Response
Dear All reviewers/Editor & Ms. Catherine Yuan,
Thanks for your e-mail dated at Aug. 25, 2023.
Enclosed please find manuscript files of the paper entitled above, by T.-Z. Li et.al. In the revised
one (R1micromachines-2535924Revised.doc), all the comments of the reviewers have been
overcome and marked with bold-highlight words. Hope this revised one can be accepted to publish on the micromachines journal (MDPI).
In addition, the special revisions per reviewers’ comments (AnswerQueries(micromachines 2535924-R2).pdf) has been attached for editors’ and reviewers’ convenience to check Your kind
assistance in dealing with this matter is my most appreciated.
Best regards from Sincerely Yours,
Jyh-Jier HO, Ph.D. 28 Aug., 2023

Reviewer 4 Report
In the original manuscript, I feel the author’s attitude for this work is very bad. There are many mistakes for this work. Thus, I give a rejection suggestion. In the revised manuscript, the author has corrected it. However, the author mentioned in the reply letter, the power conversion efficiency (PCE) over-time measurement of the PV modules are quite useful and important data to further understanding stability. I do not understand why the author can not provide this data and just mentioned keeping in mind on the next-step research in future. This is a very important data. If the author can not provide it, I still suggest to rejection.
Author Response

(The authors gave the same response as above.)

Round 3
Reviewer 4 Report
The author has solved all my questions.